# Solvent Effects on Gelation Behavior of the Organogelator Based on L-Phenylalanine Dihydrazide Derivatives

**DOI:** 10.3390/ma12121890

**Published:** 2019-06-12

**Authors:** Yang Yu, Ning Chu, Qiaode Pan, Miaomiao Zhou, Sheng Qiao, Yanan Zhao, Chuansheng Wang, Xiangyun Li

**Affiliations:** 1The Key Laboratory of the Inorganic Molecule-Based Chemistry of Liaoning Province and Laboratory of Coordination Chemistry, Shenyang University of Chemical Technology, Shenyang 110142, China; chuning0202@163.com (N.C.); pangqiaode89@163.com (Q.P.); dzjzzhoumiao@163.com (M.Z.); 18640104797@163.com (S.Q.); yanan1116@163.com (Y.Z.); wchsh18@163.com (C.W.); 2Yingkou Baoshan Ecology Coating Co., Ltd., Yingkou 115004, China

**Keywords:** supramolecular, organogelator, self-assemble, sol–gel, L-phenylalanine

## Abstract

A series of organogelators based on L-phenylalanine has been synthesized and their gelation properties in various organic solvents were investigated. The results showed that these organogelators were capable of forming stable thermal and reversible organogels in various organic solvents at low concentrations, and the critical gel concentration (CGC) of certain solvents was less than 1.0 wt%. Afterward, the corresponding enthalpies (Δ*H*_g_) were extracted by using the van ’t Hoff equation, as the gel–sol temperature (*T*_GS_) was the function of the gelator concentration. The study of gelling behaviors suggested that L-phenylalanine dihydrazide derivatives were excellent gelators in solvents, especially BOC–Phe–OdHz (compound 4). The effects of the solvent on the self-assembly of gelators were analyzed by the Kamlet–Taft model, and the gelation ability of compound 4 in a certain organic solvent was described by Hansen solubility parameters and a Teas plot. Morphological investigation proved that the L-phenylalanine dihydrazide derivatives could assemble themselves into an ordered structure such as a fiber or sheet. Fourier-transform infrared spectroscopy (FTIR) and hydrogen nuclear magnetic resonance (^1^H NMR) studies indicated that hydrogen bonding, π–π stacking, and van der Waals forces played important roles in the formation of a gel.

## 1. Introduction

Low molecular weight organogelators (LMOGs) are attractive because of their reliability with regard to the formation of organogels and self-assembled fibrillar networks [1]. The organogels, which are held together by non-covalent interactions, can be transformed to a solution phase by other physical and chemical stimuli [2,3,4]. Since organogels respond quickly when they are stimulated by external environmental factors, many recent studies have shown that they can be found in applications in various fields such as drug release [5], oil spill recovery [6], mesoporous materials [7], photoresponsive materials [8], nanostructure templates [9], drug delivery and tissue engineering [10], and the detection of heavy metal ions in water [11].

The formation of physical supramolecular gels arises from non-covalent interactions including hydrogen bonding, π–π stacking, coordination bonds, and van der Waals forces [12,13,14,15,16,17,18,19]. All these forces can cause the formation of supramolecular aggregates. The morphologies of these aggregates are various, for example, fibers, strands, and tapes. The 3D networks are formed by supramolecular aggregates which are entangled with each other at junction zones. Therefore, the solvent molecules can be contained within the 3D network. However, it is still difficult to predict the form by only using knowledge of the structural characteristics of a candidate molecule, and it is even more difficult to predict which solvents could be formed into gels [20]. Nonetheless, it was suggested that various groups are effective for the construction of gelator molecules. Bhattacharya et al. reported the gelation process of a series of organic solvents by *N*-lauroyl-L-alanine ipamphiphiles [21]. They showed that an amide H-bond formed between two nearby molecules was the driving force for the gelation process. Certain studies have shown that hydrogen-bonding units usually use the amide and urea groups, and the aromatic rings are commonly applied to introduce π–π stacking interactions. Alkyl chains always function as the hydrophobic groups [22]. Because of the chiral centers supplied by amino acids, they are frequently used as building units to form gelators. It has been reported that certain types of gelators have been modified by amino acids [23,24,25,26,27]. The relationship between solvent and gelator has been observed, and solvent parameters were used to study the interactions between the solvent and gelators.

In this research, a series of L-phenylalanine dihydrazide derivatives (compounds 1, 2, 3, and 4) were designed and synthesized. The organogelators were based on L-phenylalaniney, t-butyloxy carbonyl (BOC) and long alkyl chain units that were incorporated into the structure. The gelling behaviors of the compounds in various solvents were studied. Furthermore, the effect of solvents and the length of the alkyl chains on the gelation characteristics were investigated. The morphology of organogels and the bonding information of compound 4 were studied by scanning electron microscope (SEM), FTIR, and ^1^H NMR.

## 2. Materials and Methods

### 2.1. Materials

All of the starting compounds (methyl L-phenylalaninate hydrochloride (H–Phe–OmeCl), BOC, n-dodecyl chloride, n-tetradecanoyl chloride, n-hexadecanoyl chloride, and n-octadecanoyl chloride) were obtained from Yangzhou Baosheng Biochemical Co., Ltd. Hydrazine hydrate was purchased from Tianjin Bodi Chemical Co., Ltd. The solvents (such as methanol, alcohol, toluene, chlorobenzene) were obtained from Sinopharm Chemical Reagent Co., Ltd., and were of high quality, which allowed them to be applied without further purification steps. Some other materials were processed beforehand (purified, dried, or freshly distilled) to meet the experimental requirements.

### 2.2. Synthesis

The target compounds were synthesized according to Scheme 1. The organogelator was easily synthesized in line with the previously reported [25]. In the first step, H–Phe–OMeCl (0.03 mol) was dissolved in water (100 mL), and Na_2_CO_3_ (6.36 g) was added under stirring. Then, BOC (0.03 mol) was added into the mixed solution slowly and stirred for 14 h. The aqueous layer was extracted three times with EtOAc, then the combined organic layers were dried using anhydrous MgSO_4_ and concentrated by reduced pressure distillation to obtain BOC–Phe, which was a colorless and transparent oily liquid.

Dissolved BOC–Phe in CH_3_OH (100 mL) then NH_2_NH_2_·H_2_O (0.15 mol) were added dropwise into the mixture and stirred for 2 h at room temperature. The solvent was removed under reduced pressure and dissolved in CHCl_3_. The solution was thoroughly washed with NaCl solution, and dried using anhydrous MgSO_4_. The solvent was removed under reduced pressure after filtration to obtain BOC–Phe–Hz, which was in the form of a thick translucent liquid.

A mixture of BOC–Phe–Hz (5 g) and fatty acid chloride in CH_2_Cl_2_ (100 mL) was stirred for 6 h at room temperature. Then, the solvent was removed under reduced pressure to obtain BOC–Phe–OdHz as a white solid. The final products, BOC–Phe–LdHz (compound 1), BOC–Phe–MdHz (compound 2), BOC–Phe–HdHz (compound 3), and BOC–Phe–OdHz (compound 4) were obtained and recrystallized using CH_3_CH_2_OH/EtOAc three times.

The final products were confirmed by ^1^H NMR and FTIR, the results of which are as follows:

Compound 1: FTIR (KBr, ν, cm^−1^): 3300 (ν_N-H_, amide), 3242 (ν_N-H_, amide II), 3037 (C–H, benzene), 2916 (C–H, methyl), 2850 (C–H, methyl), 1697 (C=O, urethane), 1606 (C=O, amide I), 1537 (N–H, amide II), 1475 (benzene skeleton vibration), 698 (flexural vibration of mono-substituted benzene). ^1^H NMR (500 MHz, DMSO): δ = 9.98 (s, 1H, NH), 9.83 (s, 1H, NH), 7.29 (dt, J = 14.9Hz, 5H; Ar–H), 7.20 (t, J = 7.1 Hz, 1H, NH), 4.31–4.11 (m, 1H, CH), 2.88 (d d d, J = 24.6, 13.8, 7.2Hz, 2H, Ar–CH_2_), 1.58–1.46 (m, 2H, CH_2_), 1.29 (s, 9H, (CH_3_)_3_), 1.28–1.14 (s, 18H, (CH_2_)_9_), 0.87 (t, J = 6.9 Hz, 3H, CH_3_). LC–MS: *m*/*z* 460.3 [M]. The melting temperature range was from 135 to 137 °C. 

Compound 2: FTIR (KBr, ν, cm^−1^): 3306 (ν_N-H_, amide), 3264 (ν_N-H_, amide II), 3044 (C-H, benzene), 2926 (C–H, methyl), 2850 (C–H, methyl), 1685 (C=O, urethane), 1601 (C=O, amide I), 1542 (N–H, amide II), 1466 (benzene skeleton vibration), 700 (flexural vibration of mono-substituted benzene). ^1^H NMR (500 MHz, DMSO): δ = 9.98 (s, 1H, NH), 9.82 (s, 1H, NH), 7.29 (d t, J = 15.0, 7.4Hz, 5H, Ar–H), 7.20 (s, J = 7.2Hz, 1H, NH), 4.29–4.12 (m, 1H,), 2.76 (d d, J = 13.7, 11.0 Hz, 2H, Ar–CH_2_), 1.51 (d, J = 6.7 Hz, 2H, CH_2_), 1.29 (s, 9H, (CH_3_)_3_), 1.23 (s, 22H, (CH_2_)_11_), 0.86 (t, J = 7.1Hz, 3H, CH_3_). LC–MS: *m*/*z* 488.35 [M]. The melting temperature range was from 116 to 118 °C. 

Compound 3: FTIR (KBr, v, cm^−1^): 3302 (ν_N-H_, amide), 3253 (ν_N-H_, amide II), 3037 (C–H, benzene), 2915 (C–H, methyl), 2849 (C–H, methyl), 1696 (C=O, urethane), 1677 (C=O, urethane), 1608 (C=O, amide I), 1529 (N–H, amide II). ^1^H NMR (500 MHz, DMSO): δ = 9.97 (s, 1H, NH), 9.82 (s, 1H, NH), 7.34–7.16 (m, 5H, Ar–H), 6.94 (d, J = 8.7 Hz, 1H, NH), 4.22 (t, J = 9.8 Hz, 1H, CH), 2.99 (d, J = 18.4 Hz, 1H, Ar–CH_2_), 2.86–2.65 (m, 1H, Ar–CH_2_), 2.11 (t, J = 7.3Hz, 2H, CH_2_), 1.58–1.44 (m, 2H, CH_2_), 1.30 (d, J = 17.6Hz, 9H, (CH_3_)_3_), 1.26–1.18 (m, 24H, (CH_2_)_1__2_), 0.85 (t, J = 6.8Hz, 3H, CH_3_). LC–MS: *m*/*z* 516.3 [M]. The melting temperature range was from 112 to 114 °C. 

Compound 4: FTIR (KBr, ν, cm^−1^): 3315 (ν_N-H_, amide), 3235 (ν_N-H_, amide II), 3038 (C–H, benzene), 2917 (C–H, methyl), 2854 (C–H, methyl), 1691 (C=O, urethane), 1609 (C=O, amide I), 1526 (N–H, amide II), 1481 (benzene skeleton vibration), 1374 (flexural vibration of methy-lene), 1177 (stretching vibration of C–O), 707 (flexural vibration of mono-substituted benzene). ^1^H NMR (500 MHz, DMSO): δ = 9.91 (s, 1H, NH), 9.79 (d, J = 30.5Hz, 1H, NH), 7.26 (d, J = 4.6Hz, 5H, Ar–H), 7.19 (s, 1H, NH), 4.24–4.18 (m, 1H, CH), 2.76 (d d, J = 23.0, 9.8 Hz, 2H, Ar–CH_2_), 1.56–1.47 (m, 2H, CH_2_), 1.28 (s, 9H, (CH_3_)_3_), 1.23 (s, 30H, (CH_2_)_15_), 0.84 (t, J = 6.8Hz, 3H, CH_3_). LC–MS: *m*/*z* 545.0 [M]. The melting temperature range was from 112 to 114 °C. 

### 2.3. Measurements

Melting points were determined using a Fukai X–4 digital melting point detector (Beijing Fukai Instrument Co. Ltd., Beijing, China). The heating speed was 0.5 °C/min. Infrared spectra were measured using a Nicolet Nexus 470 FTIR spectrometer (Themo Scientific Ltd., Waltham, MA, USA). The wave number was from 400 to 4000 cm^−1^. ^1^H NMR data were recorded on a Bruker ADVANCE 500 MHz spectrometer (Bruker Co., Hamburg, Germany). LC–MS analysis was performed on an AGILENT 1260 (Agilent Technologies Inc., Santa Clara, CA, USA) and Bruker Daltonics SolariX 7.0T (Bruker Daltonics Inc., Billerica, MA, USA). The size and morphology of the xerogels were obtained using a JEOL JSM-6360LV SEM (JEOL Ltd., Tokyo, Japan). XRD patterns were performed by using a Bruker D8 AVANCE instrument (Bruker AXS GmbH, Karlsruhe, Germany). CuKα from 2° to 40° (2*θ*) were recorded in steps of 0.05°.

## 3. Results and Discussion

### 3.1. Gelation Behaviors in Some Solvents

CGC (critical gelation concentration) and gelation tests of organogelators (compounds 1, 2, 3, and 4) were performed using a variety of solvents detailed in Table 1. Most of the gels were thermo-reversible and stable for a few months at a normal temperature. It proved that L-phenylalanine dihydrazide derivatives act as versatile gelators for organic solvents. 

Because of the lack of an aliphatic chain, compound BOC–Phe–Hz could only be gelated in aromatic solvents via hydrogen bonding and π–π stacking. The excellent gelation ability of gelators (especially compounds 2, 3, and 4) could be due to their long alkyl chains. Amide groups provided the hydrogen-bonding sites, benzene groups facilitated π–π stacking interactions, and long alkyl chains supplied hydrophobic interactions, enabling gelators to self-assemble in a particular organic solvent. In Table 1, compound 4 proved to possess the highest gelation ability for the most solvents among the four gelators. With the increase in the length of the alkyl chain, the CGC of the gelators was reduced and the states of gels were transformed from transparent gel (TG) to opaque gel (OG) in ester solvents. It could be concluded that a longer alkyl chain was more beneficial for self-assembly. Evidently, aromatic solvents tend to form transparent gels at low CGC. Compound 4 did not dissolve at all in certain organic solvents (e.g., n-octane and n-hexane), even upon heating to the boiling point of the solvents, due to low polarity. Typically, the gelation process depends on the relationship between gelator and solvent, which is governed by both molecular polarity and structure. 

### 3.2. Gel Stability Research

In the research, we investigated the thermal stabilities of gels formed by compound 4 using the “dropping ball” method [28,29]. The gel–sol phase-change temperatures (*T*_GS_) of compound 4 in aromatics, alcohols, and esters were increased with the increasing concentration of the gelators (see Figure 1a,c,e). It was evident that the status of the gels was mainly dependent on strong inter-molecular interactions. Thermodynamic information was extracted from these thermal experiments using the van ‘t Hoff equation (Equation (1)) [30,31]. The enthalpy (Δ*H*_g_) could be determined by plotting the relationship between the corresponding concentration (*lnC*) and the gel–sol transition temperature [(*T*_GS_)^−1^], as shown in Figure 1b,d,f. Table 2 summarizes the enthalpy values in specific solvents.
(1)$$\frac{d\ln C}{d\left(T_{\mathrm{GS}}\right)}=-\frac{\Delta H_{g}}{R}$$

Using the Kamlet–Taft model, the hydrogen-bonding effect of solvents was found to be more precisely related to the gelation ability [32,33]. The Kamlet–Taft parameters [34] (polarity parameter (π*), hydrogen-bonding donating capacity (α), and accepting capacity (β)) of some related solvents are shown in Table 2. It appears that the hydrogen-bonding donating capacity (α) and accepting capacity (β) have an effect on gel formation. However, the polarity parameter π* demonstrates the most important role. Solvents with larger π* (nitrobenzene, methyl alcohol) hold a small gel–sol transition enthalpy (Δ*H*_g_). Furthermore, the solvents with small β values (toluene, o-xylene, m-xylene, benzene, chlorobenzene, and nitrobenzene) are associated with a large Δ*H*_g_. By comparison with other aromatics, the nitrobenzene solvents with both high π* and low β values had significantly smaller Δ*H*_g_ values. 

### 3.3. Hansen Solubility Parameter and Teas plot Presentation 

Hansen solubility parameters (HSPs) have been commonly used to select solvents to dissolve polymers [34,35]. However, this requires the use of software that allows for the generation of 3D plots. On the other hand, a 2D representation (called a Teas plot) can be used to compensate for the limitations in the graphical representations of HSPs by several studies. Here, we described the gelling ability of organogelators by HSPs and a Teas plot in various kinds of organic solvents. 

The HSPs approach described molecular interactions that were categorized into three groups: dispersive interactions (*δ*_D_), polar interactions (*δ*_P_), and hydrogen bonds (*δ*_H_) (Equation (2)). The values of solvents could be found in the literature [36], and the states of the organogelators were determined by testing the gelation ability in various kinds of organic solvents. The gelation ability of organogelators could be characterized by the modified distance *R* between the solvent (*δ*_D_, *δ*_P_, *δ*_H_) and solute (*δ^′^*_D_, *δ^′^*_P_, *δ^′^*_H_) species (Equation (3)):(2)$$\delta =\sqrt{{\left(\delta_{D}\right)}^{2}+{\left(\delta_{P}\right)}^{2}+{\left(\delta_{H}\right)}^{2}},$$
(3)$$R=\sqrt{4{\left(\delta_{D}-\delta_{D}^{\prime }\right)}^{2}+{\left(\delta_{P}-\delta_{P}^{\prime }\right)}^{2}+{\left(\delta_{H}-\delta_{H}^{\prime }\right)}^{2}},$$

The HSPs were normalized in a triangular 2D representation with coordinates determined by Equation (4):(4)$$f_{i}=\frac{\delta_{I}}{\delta_{D}+\delta_{P}+\delta_{H}}\;i=d,\;p\;\mathrm{and}\;h;I=D,\;P\;\mathrm{and}\;H,$$

L-phenylalanine dihydrazide derivatives with four different alkyl chain lengths self-assembled in certain organic solvents under a low CGC (≤3.0 wt%) to form gels. Figure 2 displays the gelation data of organogelators, which were represented in Hansen space (Figure 2a,c,e,g) and Teas plot (Figure 2b,d,f,h) presentations. Evidently, any two points close to each other in the 3D HSPs were also close to each other in the Teas plot representation. The transparent gels (TG) or opaque gels (OG) were marked by red or black spheres, and the center of the gelation sphere was depicted in gray. The cyan spheres indicated that the gels could not form at low concentrations. However, the gels could be formed after the concentration of the gelators was increased (>3.0 wt%) in this case. The gelation behaviors of compounds 1, 2, 3, and 4 showed that the gels could be formed if HSPs of the solvents were situated in the range 16.8 ≤ *δ* ≤ 29.6 MPa^1/2^. After calculation, the gelation of four gelators could be achieved if HSPs were situated in the center of the sphere (the coordinates of the center were 17.4, 8.4, 11.3). MPa^1/2^ and radius R ≈ 14.8 MPa^1/2^. The modified distance R values for compounds 1, 2, 3, and 4 were similar, and the CGC of organogelators was reduced as the alkyl chain length was increased. For organic solvents with low polarities, such as n-octane, n-hexane, and cyclohexane, compounds 1 and 2 were not completely dissolved. Compounds 3 and 4 could form gels in cyclohexane, but were still hard to dissolve in the other two solvents, which indicated that the polarity of the organic solvent and the alkyl chain length both played important roles in the gelation process. Compound 4 demonstrated the best gelling ability with the lowest CGC values among the four gelators in most organic solvents. 

### 3.4. Morphology Studies of the Gels

Compound 4 was dissolved in organic solvents, such as alcohols, esters, and aromatic solvents to form organogels. The morphology of Compound 4 xerogel in cyclohexane and in aromatic solvent was studied by SEM (Figure 3). Because the self-assembly process in the solvents differed, the aggregators showed completely different morphologies. In cyclohexane, benzene, nitrobenzene, and p-xylene, the structures of xerogels were fiber-like. The diameter of these fibers was between dozens and hundreds of nanometers (Figure 3a,b,e,h). Compared with the xerogel in benzene, the xerogel in cyclohexane showed a well-dispersed 3D network structure. This could be explained by the weak ability to attract electrons and strong steric hindrance of the cyclohexane molecule during the self-assembly process. The micrographs of xerogel in nitrobenzene assembled thick fibrous aggregates via non-covalent interactions and, eventually, formed irregular 3D networks. As shown in Figure 3c,d,f,g, the xerogel formed a complex sheet structure composed of numerous bumps and grooves of different scales. The xerogels in chlorobenzene and in n-butanol were irregularly arranged in mini-size, which probably was due to the low molecular polarity of the solvent.

The morphology of compound 4 xerogels was obtained from methyl alcohol, n-propyl alcohol, n-butanol, iso-butanol, 1-pentanol, and 1-hexanol at different concentrations, and the microtopographs of xerogels formed by 2.0 wt% compound 4 in n-propyl acetate, isobutyl acetate, and amyl acetate were investigated. Appendix A were presented in the Appendix A. According to previous work, the transmittance of the xerogel in certain solvents was different because of the irregularity of the gels and the molecular polarity and aromaticity of the solvent [37]. On account of the delicate interaction between the solvent and the gelator, the gelator assembled itself into an isotropous fiber or flocks structure.

### 3.5. Spectral Studies of the Gels

The structure of organogels was formed via hydrogen bonding and van der Waals forces [37], and the driving forces for gelation were evaluated by FTIR spectra [31,32,33]. The FTIR spectra of the xerogel, gel, and solution of compound 4 in CCl_4_ are shown in Figure 4. The IR spectrum of CCl_4_ gel (Figure 4b) shows the characteristic absorption peaks of H-bonded amide groups at 3306, 3231 (amide A), 1606 (amide I), and 1525 cm^−1^ (amide II). The characteristic infrared group frequency of the urethane group appears at 1697 and 1672 cm^−1^ in CCl_4_ xerogel, and 1698 and 1678 cm^−1^ in CCl_4_ gel. The characteristic infrared group frequency in CCl_4_ solution (Figure 4c), arising from non-hydrogen bonded amide groups, is at 3447 (ν_N-H_, amide A), 1608 (δ_N-H_, amide I), and 1528 cm^−1^ (amide II). The hydrogen-bonding between the amide groups and urethane groups causes a low-frequency shift in IR spectra [35,38]. The absorption bands of alkyl chains in CCl_4_ gel and antisymmetric and symmetric stretching vibrations appear at higher wave numbers than those of compound 4 in CCl_4_ xerogel or in CCl_4_ solution. It is suggested that compound 4 self-assembled into a nanostructure via inter molecular hydrogen-bonding [39,40]. Therefore, the CCl_4_ molecules were immobilized in the networks. The similar shape of IR spectra confirmed that the intermolecular structure was retained in the transition from CCl_4_ gel to xerogel. Thus, the hydrogen bonding was the major factor contributing to the formation of organogel. 

FTIR spectra of compound 4 xerogel in benzene, n-propanol, CCl_4_, and propyl acetate, respectively, are shown in Figure 5. The positions of the peaks were similar, which meant that the gels’ structures were not broken by the intermolecular interaction between the gelator and solvent. This result indicated that the driving force of organogel was hydrogen-bonding.

The driving forces of the gelation are the non-covalent bonds formed by intermolecular interactions, and the credible data of these weak interactions can be obtained from the NMR studies [41]. The self-assembly behavior of compound 4 in certain solvents was further confirmed by ^1^H NMR spectroscopy, as shown in Figure 6. 

In the experiment, as the concentration of the gelator in benzene was increased from 20 to 30 mg/mL, the chemical shifts of the aromatic protons on the alkyl chain and the internal benzene ring moved upfield. The chemical shift of the hydrogen atom in benzene and aliphatic chain indicated that π–π stacking and a hydrophobic interaction played important roles in gel-forming.

### 3.6. X-ray Diffraction of the Gels 

Figure 7 shows the regular X-ray diffraction pattern of compound 4 xerogel in benzene (4.0 wt%). The sharp peaks demonstrated that the xerogel formed a crystalline structure during gelation. 

The results indicated that due to the intermolecular interactions, the gelators rearranged and formed an orderly network structure during the self-assembly process. The low-angle X-ray diffraction pattern of compound 4 xerogel in benzene shows periodical diffraction peaks, which indicates compound 4 self-assembles into an ordered supramolecular structure. The XRD pattern shows the that the largest Bragg distance was 2.21 nm, which was consistent with the length (2.213 nm) reported for the optimal model of a single compound 4 molecule, which was measured by HyperChem 8.0 using the semi-empirical method INDO and the Polak–Ribiere algorithm (based on the data of XRD). The molecules were docked by hydrogen bonding, and the simulation diagram of hydrogen bonding is shown in Figure 8a. According to results of FTIR and low-angle XRD, it could be confirmed that long alkyl chains of compound 4 molecules formed a fully expanded interdigitated layer structure during the self-assembly process. The possible stacking model is shown in Figure 8b. However, the processing to form a gel is very complex. The interaction between the gelators and the solvent molecules should be considered. Therefore, the simulation results are limited.

## 4. Conclusions

In summary, a class of organogelators based on L-phenylalanine were synthesized and their gelation properties in various organic solvents were investigated thoroughly. According to the studies of gelling behaviors L-phenylalanine dihydrazide derivatives are great gelators among various organic solvents with thermo-reversible and thermal stabilities. The solvent effects on the self-assembly of gelators were analyzed by applying the Kamlet–Taft model, and the gelation ability of compound 4 was described by Hansen solubility parameters and a Teas plot. The gelation ability of organogelators depends on the long alkyl chain and dihydrazide moiety. These simple L-phenylalanine dihydrazide derivatives can assemble themselves into ordered aggregates such as fiber or lamellar nanostructures and form 3D networks. Morphological investigations proved the simple L-phenylalanine dihydrazide derivatives could self-assemble into an ordered structure (such as fibber or lamellar nanostructures). The results of FTIR and ^1^H NMR spectroscopy investigations indicated that hydrogen bonding, π–π stacking, and hydrophobic interactions played important roles in forming a self-assembled gel.

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
