# Peer review of "Solvent Effects on Gelation Behavior of the Organogelator Based on L-Phenylalanine Dihydrazide Derivatives"

_materials, 2019, doi:10.3390/ma12121890_

Round 1

Reviewer 1 Report

The authors report a series of four phenylalanine derivatives with supramolecular gel formation capabilities. The properties of the gels have been investigated with a myriad of conventional techniques and gelation abilities have been attempted to correlate with Kamlet-Taft and Hansen Solubility Parameters. Regrettably, the article suffers from severe neglicence (especially with respect of the literature references, i.e. on row 53: [23-267] - there are only 41 references listed in the List of References at the end of the article) and numerous solecisms requiring complete editing of the English language and style.

The Introduction is extremely short and narrow, and contains defects and inaccuracies. For example, on row 30: "Organogelators with low molecular (LMOGs) are...", whereas the correct terminology with the abbreviation would be "low molecular wight organogelatos (LMWOGs) are..." and on row 38: "The formation of gels arises from non-covalent interactions...", which only applies for physical aka supramolecular gels but definitely not for chemical aka polymer gels. 

In the experimental section the vendors of the chemicals and solvents should be specified. Also the detailed methods of "processing of some other materials beforehand to meet the experimental requirement" should be given. The NMR analysis of the compounds is extremely confusing - it is suggested that the authors will number the protons and carbons in each of the compounds in order to enable unambiguous assignments of the resonance signals. I think there's also some errors in the analysis of the coupling patterns, e.g. the resonant signal of the amide proton is marked as a triplet, whereas it should appear as a doublet (3JHH between the amide proton and only one alpha proton), the aromatic protons are marked to give a triplet of doublets instead of the three magnetically inequivalent aromatic protons each of which giving their own chemical shifts and coupling patterns (partially overlapping for sure), etc. Thus, the NMR analysis must be carefully revised for all of the compounds. Additionally, 13C and 1H,1H COSY, 1H,13C HSQC, and 1H,13C HMBC measurements should be performed for unambiguous spectral assignments. Also the names of the instruments should be revised - instead of Bruker Advance or Aglient the names of the instruments should be written as Bruker AVANCE and AGILENT.

The chapter 'Results and Discussion' is written in a declaratory manner. Not many conclusions have been drawn based on the obtained results. When correlating the Kamlet Taft parameters with the gelation tendencies, the authors do not give any kind of hypothesis of why the other aromatic solvents with low beta values possess high sol-gel transition enthalpies, whereas nitrobenzene with high polarity parameter and low beta value possesses low enthalpy? Moreover, the question "How are the enthalpies and gelation tendencies correlated and why?" remains unanswered. The literature reference for the Kamlet-Taft parameter values should be given.

The idea of inspecting the Hansen Solubility Parameters and Teas plots with respect of the gelation abilities of the gelators in different solvents is very interesting. However, a much more detailed analysis and interpretation of the obtaines results remains missing. The authors should explain how the use of Kamlet-Taft parameters and/or HSPs could be used in designing gelators and/or various gel systems.

Since the SEM micrographs don't provide any groundbraking information, it is enough to present only one Figure representing the different morphologies, e.g. with the micrographs representing the fibrillar, sheet-like, and flake-like morphologies. The other micrographs can be presented in supporting information.

In Figure 6 the most important spectrum, the one of the actual gel, is missing. Also in Figure 7 one of the spectra (xerogel obtained from CCl4) is lacking. On rows 272-273 the authors claim that alkyl groups of compound 4 self-assemble into nanostructures via intermolecular hydrogen bonding. Traditionally, hydrogen bond formation requires an acceptor and a donor, neither of which appear in hydrocarbon chains. Thus the conclusion is highly questionnable.

As in the chapter 'Experimental' the erroneous terminology and incorrect interpretations characterize the conclusions made on the basis of the measured the NMR spectra. First of all, NMR spectra consist of resonance peaks of the nuclei in question, not absorption peaks. Second, the intensity of the peak is not the quantity of the interest. What is constantly examined in 1H NMR spectra are the integrals, i.e. the areas under the NMR resonance lines. The area is proportional to the number of hydrogens, and by integrating the spectrum one can obtain information regarding the relative numbers of chemically distinct hydrogens. Furthermore, if the resonant signals move to the right in the NMR spectrum, as is the case in the current study upon increasing the concentration, the signals move to the higher field region (upfield), NOT to the lower field region as the authors state. In the case of an intermolecular hydrogen bonding, the acceptor protons should actually move to higher resonant frequency values (downfield, see. e.g. H. Günther, NMR Spectroscopy- Basic Principles, Concepts, and Applications in Chemistry; Wiley, 2013), not to lower chemical shift values (upfield), as happens in the current study (Figure 8). Based on the 1H NMR spectra one can thus not say that intermolecular hydrogen bonds drive the gel formation. Further, the reasoning for the existence of pi-pi and/or hydrophobic interactions based on the NMR spectra is weak. It is obvious from the text that the authors have severe deficiencies in their NMR knowledge, and they should thus carefully study the basics on NMR theory before drawing any conclusions based on the results.

Even though the XRD suggest a highly organized structure, there is always the question whether a xerogel really represents the native gel with the solvent. Care must be taken in drawing any conclusions on native gels based solely on the results obtained for the xerogels. The calculated model for the packing of the molecules seems reasonable, but it too lacks the solvent molecules that form the majority of the gel (97 % w/v). Their influence can not be igonred in theoretical calculations. 

In the current manuscript the authors present a series of simple low molecular weight organogelators based on phenylalanine, whose gelation tendencies have been examined and attempts on correlating the gelation with the Kamlet-Taft and Hansen Solubility Parameters have been done. The structures of the gels have been investigated by SEM, XRD, FT-IR, and NMR spectroscopy. Unfortunately, the analyses remain incomplete and the conclusions are drawn without reasonable and sufficient results. 

Author Response

Point 1: Comments and Suggestions for Authors

The authors report a series of four phenylalanine derivatives with supramolecular gel formation capabilities. The properties of the gels have been investigated with a myriad of conventional techniques and gelation abilities have been attempted to correlate with Kamlet-Taft and Hansen Solubility Parameters. Regrettably, the article suffers from severe neglicence (especially with respect of the literature references, i.e. on row 53: [23-267] - there are only 41 references listed in the List of References at the end of the article) and numerous solecisms requiring complete editing of the English language and style.

Response 1: I’m very sorry for the error about the reference number and I have revised all the mistakes.

Point 2:The Introduction is extremely short and narrow, and contains defects and inaccuracies. For example, on row 30: "Organogelators with low molecular (LMOGs) are...", whereas the correct terminology with the abbreviation would be "low molecular wight organogelatos (LMWOGs) are..." and on row 38: "The formation of gels arises from non-covalent interactions...", which only applies for physical aka supramolecular gels but definitely not for chemical aka polymer gels. 

Response 2: I have already revised them.

Point 3::In the experimental section the vendors of the chemicals and solvents should be specified. Also the detailed methods of "processing of some other materials beforehand to meet the experimental requirement" should be given. The NMR analysis of the compounds is extremely confusing - it is suggested that the authors will number the protons and carbons in each of the compounds in order to enable unambiguous assignments of the resonance signals. I think there's also some errors in the analysis of the coupling patterns, e.g. the resonant signal of the amide proton is marked as a triplet, whereas it should appear as a doublet (3JHH between the amide proton and only one alpha proton), the aromatic protons are marked to give a triplet of doublets instead of the three magnetically inequivalent aromatic protons each of which giving their own chemical shifts and coupling patterns (partially overlapping for sure), etc. Thus, the NMR analysis must be carefully revised for all of the compounds. Additionally, 13C and 1H,1H COSY, 1H,13C HSQC, and 1H,13C HMBC measurements should be performed for unambiguous spectral assignments. Also the names of the instruments should be revised - instead of Bruker Advance or Aglient the names of the instruments should be written as Bruker AVANCE and AGILENT.

Response 3: All the vendors of the chemicals and solvents have been specified. The processing the materials beforehand have been given. And the names of the instruments have been revised.

It is very sorry that we couldn’t offer the spectrogram of 13C and 1H,1H COSY, 1H,13C HSQC, and 1H,13C HMBC of the compound within a short time.

The NMR analysis has revised for all of the compounds.

Point 4:The chapter 'Results and Discussion' is written in a declaratory manner. Not many conclusions have been drawn based on the obtained results. When correlating the Kamlet Taft parameters with the gelation tendencies, the authors do not give any kind of hypothesis of why the other aromatic solvents with low beta values possess high sol-gel transition enthalpies, whereas nitrobenzene with high polarity parameter and low beta value possesses low enthalpy? Moreover, the question "How are the enthalpies and gelation tendencies correlated and why?" remains unanswered. The literature reference for the Kamlet-Taft parameter values should be given.

Response 4: In general, the stronger the hydrogen bonding capacity of solvent, the easier the formation of hydrogen bond between gelator and solvent, and the worse the stability of gel. (e.g. Edwards, W.; Lagadec, C. A.; Smith, D. K. Soft Matter 2011, 7(1), 110. dio:10.1039/c0sm00843e.) Nitrobenzene with high polarity parameter and low beta value possesses low enthalpy and aromatic solvents with low beta values possess high sol-gel transition enthalpies.

The literature reference for the Kamlet-Taft parameter values has been given, and the number of the reference is 34.

Point 5:The idea of inspecting the Hansen Solubility Parameters and Teas plots with respect of the gelation abilities of the gelators in different solvents is very interesting. However, a much more detailed analysis and interpretation of the obtaines results remains missing. The authors should explain how the use of Kamlet-Taft parameters and/or HSPs could be used in designing gelators and/or various gel systems.

Response 5: As mentioned above, in general, the stronger the hydrogen bonding capacity of solvent, the easier the formation of hydrogen bond between gelator and solvent, and the worse the stability of gel. So we could predict the stability of the gels. The gels could be formed if HSPs of the solvents were situated in the range of Hansen sphere. And similar to the HSPs, the solvent which can be gelated by gelator will cluster in a specific area in Teas plot model. So we could predict if the solvent could be gelated.

Point 6:Since the SEM micrographs don't provide any groundbraking information, it is enough to present only one Figure representing the different morphologies, e.g. with the micrographs representing the fibrillar, sheet-like, and flake-like morphologies. The other micrographs can be presented in supporting information.

Response 6: According to the suggestions, I have presented the other SEM images in SI.  

Point 7:In Figure 6 the most important spectrum, the one of the actual gel, is missing. Also in Figure 7 one of the spectra (xerogel obtained from CCl4) is lacking. On rows 272-273 the authors claim that alkyl groups of compound 4 self-assemble into nanostructures via intermolecular hydrogen bonding. Traditionally, hydrogen bond formation requires an acceptor and a donor, neither of which appear in hydrocarbon chains. Thus the conclusion is highly questionnable.

Response 7: The data was lost during the output processing, so that the spectrum was missed in Figure 6 and Figure 7. I have replaced the correct Figure and renumbered them.  

The alkyl group is not able to form hydrogen bond. It should be the amino groups of compound 4.

Point 8:As in the chapter 'Experimental' the erroneous terminology and incorrect interpretations characterize the conclusions made on the basis of the measured the NMR spectra. First of all, NMR spectra consist of resonance peaks of the nuclei in question, not absorption peaks. Second, the intensity of the peak is not the quantity of the interest. What is constantly examined in 1H NMR spectra are the integrals, i.e. the areas under the NMR resonance lines. The area is proportional to the number of hydrogens, and by integrating the spectrum one can obtain information regarding the relative numbers of chemically distinct hydrogens. Furthermore, if the resonant signals move to the right in the NMR spectrum, as is the case in the current study upon increasing the concentration, the signals move to the higher field region (upfield), NOT to the lower field region as the authors state. In the case of an intermolecular hydrogen bonding, the acceptor protons should actually move to higher resonant frequency values (downfield, see. e.g. H. Günther, NMR Spectroscopy- Basic Principles, Concepts, and Applications in Chemistry; Wiley, 2013), not to lower chemical shift values (upfield), as happens in the current study (Figure 8). Based on the 1H NMR spectra one can thus not say that intermolecular hydrogen bonds drive the gel formation. Further, the reasoning for the existence of pi-pi and/or hydrophobic interactions based on the NMR spectra is weak. It is obvious from the text that the authors have severe deficiencies in their NMR knowledge, and they should thus carefully study the basics on NMR theory before drawing any conclusions based on the results.

Response 8: Thanks for the suggestion about the NMR spectra. It’s really using the erroneous terminology. It has been revised.

In Figure 8 (The new number is Figure 6 in the revised manuscript ), the resonance peaks of the porton in aromatic, alkyl chain, benzene ring of compound 4 in benzene gel were moved to upfield according to increase the concentration. It was indicated that the hydrogen bond, π-π stacking and hydrophobic effect influence the gel forming.  

Point 9: Even though the XRD suggest a highly organized structure, there is always the question whether a xerogel really represents the native gel with the solvent. Care must be taken in drawing any conclusions on native gels based solely on the results obtained for the xerogels. The calculated model for the packing of the molecules seems reasonable, but it too lacks the solvent molecules that form the majority of the gel (97 % w/v). Their influence cannot be igonred in theoretical calculations. 

Response 9: The xerogel was prepared by freezing the gel for 2h in refrigerator and drying in a vacuum freeze-dryer. It was preserved for the self-assembly structure at the greatest extent. The interaction between gels and solvents does affect the self-assembly process. It must be paid attention.

Reviewer 2 Report

Review of Ms. Ref. No.:  materials-485168

The manuscript entitled „Solvent Effects on Gelation Behavior of the Organogelator Based on L-Phenylalanine Dihydrazide Derivatives” by  Yang Yu, Ning Chu, Qiaode Pan, Miaomiao Zhou, Sheng Qiao, Yanan Zhao,Chuansheng Wang and Xiangyun Li, describes the synthesis of a group of gelators classified as low molecular weight gelators (LMWG) and characterization of its properties in the gel phase. The authors use experimental techniques (FT-IR, NMR, X-ray diffraction, SEM), which are well chosen for characterization of basic properties of investigated systems. The performed measurements seem to be of good quality. However, the analysis of obtained data is very shallow and in some places even missing. Although the authors put a lot of effort in obtaining new gelators, (the part of a manuscript describing the synthesis is written correctly), they did not put as much in analysis of obtained experimental results from FT-IR, NMR, and TGS data. Some of the used phrases are exaggerated, for example, writing that concentration of gelator in prepared gels £ 3.0% is “extremally” low is not justified as this is rather used for concentrations £ 1.0%. The authors should consider changing this value as they describe some cases in which the concentration is £ 1.0%. Another example of lack of consistency is that the authors describe the enthalpy of the gel-sol phase transition in studied gels and use for this analysis the Von’t Hoff equation, but have not provided the equation in the manuscript. Moreover, the Van 't Hoff equation relates the change in the equilibrium constant, Keq, of a chemical reaction to the change in temperature, T, given the standard enthalpy change, ΔH, for the process, whereas the gel-sol phase transition is a physical transition related to disruption of the gel matrix by thermal energy. Thus this transition is described in the literature as a process similar to the dissolution of a crystal, for which the Schroeder-van Laar equation is used to calculate the enthalpy of the process. Although both equations are similar in mathematical form and lead to similar results, the used by authors description is not adequate. Summarizing, the authors are appreciated for taking the effort to synthesized new LWMGs, which are an important group of materials, butthe weaknesses of the presented manuscript needs to be stated out. The authors are encouraged to consider the comments and improve the paper.     

1) In abstract, the authors use the phrase „extremely low concentration” for values ≤ 3%, which in the literature of the gels made by LMWG is rather used for concentrations 1.0%. As the authors later describe a case where they obtain concentrations 1.0%, this value should be used in the abstract.

2) In paragraph 2.2, the authors report the chemical shifts for signals on NMR spectra obtained for the frequency of protons equal to 500 MHz, whereas in paragraph 2.3 they wrote that the measurements were done on the 600 MHz spectrometer. Where is the mistake?

3) In paragraph 3.1, the authors describe the gelation behavior and conclude that the long alkyl chain promotes the self-assembly on one side and distort the microstructure on the other side. As the microstructure depends on the self-assembly process, this conclusion should be revised. The authors also mention the appearance of the gels, transparent or opaque and later conclude this depends on the relation between gelator and solvent. The self-assembly process depends on many factors resulting from the structure and property of the gelator and solvent, but also on the conditions in which it is carried out. Some papers treated on how the cooling rate influence on the microstructure of such LMWG gels. The authors should consider this fact, especially that they cite author who also has studied such effect, obtaining transparent or opaque gel of the same concentration of gelator but different cooling rates.

4) On figure 1a the data point for are missing, moreover how the solid lines on this figure were obtained? Are they a mathematical fit of some kind? How it could be that a single line based on two points (fig 1a) is not going through them?

5) The equation on which basis the authors calculate the enthalpy need to be provided for the readers in the manuscript.

6) The author should explain why they use the Von’t Hoff equation instead of Schroeder van Laar equation for calculation of the gel-sol transition enthalpy. In the reviewer opinion, the Von’t Hoff equation is used for different situation, then the described in the manuscript.

7) Figure 6, the spectrum for a gel in CCl4 is missing.

8) Figure 8, the description of the figure is not adequate. What means a, b, c and d panels on the figure? Moreover, the lines should be assigned for a better understanding of the data.

9) In paragraph 3.5 under figure 8, the authors have written: „It should be noted that the highest broadening was observed following the concentration increased due to the restricted molecular motion.” Unfortunately, the authors did not write which lines were analyzed, what were the line widths and how did they change. Form the figure 8 no such observations can be done without numerical analysis of the line widths.

10) Figure 10, the authors present the simulation diagram and the optimal structure of compound 4. Unfortunately, the authors did not write in the manuscript how the simulations were done. What kind of software was used and on what level of theory the optimization was done for compound 4.

11) The manuscript has numerous errors concerning the citing papers (numbering)

12) The English level of the manuscript is very low; there are numerous grammatical and spelling mistakes. The writing of the manuscript needs to be very much improved.

The graphical abstract, which should show the most important value of the work, is missing. The authors are encouraged to make one.

Taking into the account the above comments and the current state of the manuscript I regret to inform that in present form the manuscript cannot be recommended for publication in the journal. The authors are advised to revise the manuscript, rewrite and improve the level of the performed analysis, and improve the English. After this change, which in my opinion exceeds the major revision, the manuscript can be reconsidered for publication.

Author Response

Point 1: In abstract, the authors use the phrase „extremely low concentration” for values ≤ 3%, which in the literature of the gels made by LMWG is rather used for concentrations ≤ 1.0%. As the authors later describe a case where they obtain concentrations ≤ 1.0%, this value should be used in the abstract.

Response 1: We have revised the value in the abstract. 

Point 2: In paragraph 2.2, the authors report the chemical shifts for signals on NMR spectra obtained for the frequency of protons equal to 500 MHz, whereas in paragraph 2.3 they wrote that the measurements were done on the 600 MHz spectrometer. Where is the mistake?

Response 2: Sorry for the mistake of the spectrometer frequency. 500Mz is correct, and we have modified it.

Point 3: In paragraph 3.1, the authors describe the gelation behavior and conclude that the long alkyl chain promotes the self-assembly on one side and distort the microstructure on the other side. As the microstructure depends on the self-assembly process, this conclusion should be revised. The authors also mention the appearance of the gels, transparent or opaque and later conclude this depends on the relation between gelator and solvent. The self-assembly process depends on many factors resulting from the structure and property of the gelator and solvent, but also on the conditions in which it is carried out. Some papers treated on how the cooling rate influence on the microstructure of such LMWG gels. The authors should consider this fact, especially that they cite author who also has studied such effect, obtaining transparent or opaque gel of the same concentration of gelator but different cooling rates.

Response 3: We have revised the conclusion that the long alkyl chain promotes the self-assembly.

In this paper, the cooling rates was 2℃/min, so the influence of the conditions for gel forming could remove.   

Point 4: On figure 1a the data point for are missing, moreover how the solid lines on this figure were obtained? Are they a mathematical fit of some kind? How it could be that a single line based on two points (fig 1a) is not going through them?

Response 4: The data was lost during the output processing, so that the specturm was missed. I have replaced the correct Figure and renumbered them. 

Point 5: The equation on which basis the authors calculate the enthalpy need to be provided for the readers in the manuscript.

Response 5: The equation has been added in the manuscript.

Point 6: The author should explain why they use the Von’t Hoff equation instead of Schroeder van Laar equation for calculation of the gel-sol transition enthalpy. In the reviewer opinion, the Von’t Hoff equation is used for different situation, then the described in the manuscript.

Response 6: The Van 't Hoff equation is as follow:

dlnC/d(T_GS ) =-(∆H_g)/R

The enthalpy (ΔHg) could be determined by plotting the relationship between corresponding concentration (lnC) and gel-sol transition temperature [(TGS)-1].

Point 7: Figure 6, the spectrum for a gel in CCl4 is missing.

Response 7: The data was lost during the output processing, so that the specturm was missed. I have replaced the correct Figure and renumbered them. 

Point 8: Figure 8, the description of the figure is not adequate. What means a, b, c and d panels on the figure? Moreover, the lines should be assigned for a better understanding of the data.

Response 8: In Figure 8 (the new number is Figure 6), a, b, c and d panels mean locally amplified chemical shift in order to show the chemical shift changes according to the concentration increasing.

Point 9: In paragraph 3.5 under figure 8, the authors have written: „It should be noted that the highest broadening was observed following the concentration increased due to the restricted molecular motion.” Unfortunately, the authors did not write which lines were analyzed, what were the line widths and how did they change. Form the figure 8 no such observations can be done without numerical analysis of the line widths.

Response 9: I’ m very sorry to make the confused about the NMR analysis. It was the chemical shift moved to upfield, not the line widths.  

Point 10: Figure 10, the authors present the simulation diagram and the optimal structure of compound 4. Unfortunately, the authors did not write in the manuscript how the simulations were done. What kind of software was used and on what level of theory the optimization was done for compound 4.

Response 10: The molecular simulation software is HyperChem 8.0. The semi-empirical method INDO and Polak-Ribiere algorithm were used to simulate the possible stacking model of compound 4.

Point 11: The manuscript has numerous errors concerning the citing papers (numbering)

Response 11: It’s very sorry for the errors about the number of the references. The number of the citing paper has been revised.

Point 12: The English level of the manuscript is very low; there are numerous grammatical and spelling mistakes. The writing of the manuscript needs to be very much improved.

Response 12: We revised the manuscript and corrected the grammatical and spelling mistakes carefully.

Point 13: The graphical abstract, which should show the most important value of the work, is missing. The authors are encouraged to make one.

Response 13: The graphical abstract has been upload to the submission system.

Round 2

Reviewer 1 Report

The text still suffers from severe solecisms and misspellings. Before potential publication the article must undergo extensive revision of language. Especially the Abstract, Introduction, and Experimental should be carefully revised with respect of the language. Moreover, the characterization of the synthesized compounds is insufficient. The assignments of the NMR spectra are ambiguous and in some cases erroneous. They must be revised. The specifications for the equipmentation are incorrect (the instruments are called Bruker Avance (not Advance) and Agilent (not Aglient)). 

In the Figure caption of Figure 6 subsection a) shows amide protons rather than amino protons and subsection d) methyl (CH3) protons instead of methylene (CH2) protons. Furthermore, it remains relatively unclear, how the movement of the chemical shifts of the aromatic protons to smaller chemical shifts actually proves that hydrophobic interactions play a role in the gel formation? The fact that PXRD shows the diffraction pattern for the xerogel, the gel without the solvent, and that molecular modeling is performed without the solvent molecules should be at least discussed before too straightforward conclusions of the molecular level structure of the native gels are drawn. 

Author Response

Thank alot for the comments and suggestions. We have revised the paper carefully. The respone point by point is as follow:

Point 1: The text still suffers from severe solecisms and misspellings. Before potential publication the article must undergo extensive revision of language. Especially the Abstract, Introduction, and Experimental should be carefully revised with respect of the language.

Response 1: The manuscript has been checked by a professional English editing service provided by MDPI.

Point 2: Moreover, the characterization of the synthesized compounds is insufficient. The assignments of the NMR spectra are ambiguous and in some cases erroneous. They must be revised.

Response 2: We have already revised the the assignments of the NMR spectra.

Point 3: The specifications for the equipmentation are incorrect (the instruments are called Bruker Avance (not Advance) and Agilent (not Aglient)). 

Response 3: The spelling mistakes of the specifications for the equipmentation have been revised.

Point 4: In the Figure caption of Figure 6 subsection a) shows amide protons rather than amino protons and subsection d) methyl (CH3) protons instead of methylene (CH2) protons. Furthermore, it remains relatively unclear, how the movement of the chemical shifts of the aromatic protons to smaller chemical shifts actually proves that hydrophobic interactions play a role in the gel formation?

Response 4: The Figure caption of Figure 6 subsection a) and d) have been revised. The chemical shifts of the aromatic protons on the alkyl chain and the internal benzene ring moved upfield according to the increased concentration of the gelator in benzene. That may be the shielding effects of these protons were increased. Andthen, the shielding effects of these protons were caused by the hydrophobic interactions of the long aliphatic chain.

Point 5: The fact that PXRD shows the diffraction pattern for the xerogel, the gel without the solvent, and that molecular modeling is performed without the solvent molecules should be at least discussed before too straightforward conclusions of the molecular level structure of the native gels are drawn. 

Response 5: Paragraph 3.4 discussed the morphology of compound 4 in different solvent. The SEM images was completely different. It indicated that the self-assemble process was affected by solvent.So we chosed compound 4 xerogel in benzene as a representative. 

Thanks again.

Reviewer 2 Report

Concerning the answer on comment 6

The reviewer knows how to calculate the enthalpy of the gel-sol phase transition. The point is that the authors use the Van't Hoff equation, which in its mathematical form is the same as Schroeder van Laar equation but under this name is used for chemical reactions, not for structural phase transitions. In the future, the authors are encouraged to read works from Schroeder van Laar. The description which they will find in the literature will help them in concluding of their own research.

Author Response

Point 1: The reviewer knows how to calculate the enthalpy of the gel-sol phase transition. The point is that the authors use the Van't Hoff equation, which in its mathematical form is the same as Schroeder van Laar equation but under this name is used for chemical reactions, not for structural phase transitions. In the future, the authors are encouraged to read works from Schroeder van Laar. The description which they will find in the literature will help them in concluding of their own research.

Response 1: The paper of Seo (Sang Hyuk Seo, Ji Young Chang. Organogels from 1H-imidazole amphiphiles entrapment of a hydrophilic drug into strands of the self-Assembled amphiphiles. [J]. Chem. Mater., 2005, 17(12): 3249-3254.) presents that If the sol-gel transition is comparable to melting of crystals, the sol-gel transition enthalpy can be estimated by van’t Hoff relationship. So we also use the Van't Hoff equation to calculate the enthalpy of the gel-sol phase transition.

We appreciate very much the advices of the reviewer. The remarks will benefit the improvement of the paper and our research. We will read works from Schroeder van Laar in the future.

Thanks again.
